# Effects of the Ball Milling Process on the Particle Size of Graphene Oxide and Its Application in Enhancing the Thermal Conductivity of Wood

**Na Zhang [1,2], Yiqun Mao [2], Shuangshuang Wu [2] and Wei Xu [1,2,*]**

1   Co-Innovation Center of Efficient Processing and Utilization of Forest Resources, Nanjing Forestry University, Nanjing 210037, China
2   College of Furnishings and Industrial Design, Nanjing Forestry University, Nanjing 210037, China
*   Correspondence: xuwei@njfu.edu.cn; Tel.: +86-025-8542-7469

**Abstract:** To improve the dispersion of graphene oxide particles in wood for better thermal conductivity, this paper proposes the feasibility of obtaining graphene oxide with a smaller particle size using ball milling and its application in melamine resin-modified poplar veneer. The median diameter of multilayer graphene oxide was measured to learn the effects of different ball milling conditions on the particle size of graphene oxide, and the optimum ball milling process was chosen. In addition, the microscopic characterization of graphene oxide under the optimum ball milling process was carried out to investigate the microstructural changes in multilayer graphene after ball milling. Furthermore, the thermal conductivity of the graphene oxide/melamine resin-impregnated mixture modified veneer with the optimum ball milling process was also tested. The results show that, under the optimum ball milling process conditions of SDS wet ball milling with a vibration frequency of 30 Hz for 60 min, the particle size of the multilayer graphene was the smallest, and the median diameter could be reduced to 124 nm. Simultaneously, the thermal conductivity of the melamine resin-modified poplar veneer enhanced by the ball-milled graphene reached $0.405 \ \mathrm{W \cdot m^{-1} \cdot K^{-1}}$. In addition, it revealed that the number of graphene oxide layers was reduced to four after ball milling. However, the multilayer graphene was partially oxidized, the lamellar structure was destroyed and the crystallinity was reduced.

**Keywords:** thermal conductivity; ball milling; multilayer graphene oxide; particle size; microscopic characterization

## 1. Introduction

As a renewable and versatile material, wood is widely used in construction and flooring [1]. Due to the uniform temperature distribution and slight vertical temperature differences indoors, radiant floor heating has become a popular heating method for modern buildings [2]. Compared with ceramic tiles, wooden flooring can reduce energy consumption, greenhouse gas emissions and the cost of emission reduction [3]. However, as a result of the limited thermal conductivity of wood, it is ineffective for wood to transfer heat as a floor heating material [4]. Seo et al. [5] measured the thermal and transfer performance of 21 replicates of wooden flooring materials widely used in Korea and found that laminate flooring had higher thermal conductivity than other floors. Therefore, composite laminate flooring has received more attention in recent years.

Impregnation treatment is a common method of preparing wood composite materials by immersing organic or inorganic materials into the internal pores of timber to improve wood properties [6]. In addition, melamine resins are widely used in impregnation treatments for their thermal stability. However, wood impregnated with melamine resins turns brittle; therefore, wide micro-cracks form during resin hardening. Dorieh et al. [7] found that incorporating low-level nanoscale fillers into melamine-formaldehyde resin (MFR) led

to improvements in their mechanical, physical and thermal performance and low formaldehyde emission. It has also been reported that MFR nanocomposites have the advantages of low cost, easy industrialization and high water resistance.

Because of its extremely high thermal conductivity of 2000~4000 $W \cdot m^{-1} \cdot K^{-1}$ [8], graphene and its derivatives have been applied as promising additions for enhancing MFR. It has been reported that graphene and its derivatives, such as graphite nanosheets (GNSs), could endow composites with excellent electrochemical properties and high electrical conductivity [9,10]. Moreover, Wang et al. [11] compounded GNSs with polybutene succinic acid (PBS) in order to improve the electrical conductivity and thermal stability of graphene-based nanocomposites. Celik et al. [12] added 15 vol.% graphene platelets (GPLs) to monolithic alumina at 600 °C, increasing the in-plane thermal conductivity by 44%. Wu et al. [13] explored the thermal conductivity of graphene/polyvinyl alcohol (Gr/PVA)-impregnated veneer and found that the dispersion of the graphene-impregnated modifier relates directly to its modification effect, and the dispersion can be improved by adjusting the proportions of impregnating solution and ultrasonic treatment [13]. Overall, the current studies have focused on the pretreatment of wood to enhance the effectiveness of graphene impregnation, but few studies pay attention to the preparation of graphene before impregnation.

However, graphene has a large specific surface area and interlayer van der Waals forces, which are prone to agglomerates and uneven dispersion in the matrix, thus affecting the performance of the composite material. Graphene oxide has a large number of functional groups on its surface, such as carboxyl groups, hydroxyl groups and epoxy groups. Therefore, graphene oxide has a high chemical stability and can effectively disperse the attached materials during the material compounding process [14,15]. Additionally, ball milling is an established industrial grinding technique with potential scalability that can effectively reduce the size of uniformly distributed graphite particles [16]. Bastwros et al. [17] synthesized a graphene-enhanced aluminum composite material, which improved the material's strength. The milling process did not damage the graphene and was conducive to dispersing uniformly and reducing the number of stacked layers. The strengthening was significantly affected by the dispersion of the graphene in the matrix phase. Yue et al. [18] prepared GNS-reinforced pure copper matrix composites by ball milling and hot pressing sintering and found that ball milling was beneficial to the dispersion of GNSs in the copper matrix, with good interfacial bonding, which improved the performance of the composites. In other words, ball milling reduced the particle size of the multilayered graphene powder; therefore, it could be better dispersed in the solvent and reduce the difficulty of graphene particles entering the wood. However, long-term ball milling could cause severe damage to GNSs.

Inspired by that, we used ball-milled graphene oxide mixed with melamine resin as a solution for impregnating poplar veneers. The influencing factors, the ball milling environment, the vibration frequency and the time were taken into consideration. The median diameter of multilayer graphene oxide and the thermal conductivity of ball-milled graphene oxide modified veneer were tested as the evaluation standard. Furthermore, Raman spectroscopy and X-ray diffraction (XRD) were used to characterize the microstructure change in graphene oxide in ball milling to investigate the influence of ball milling on particle size and the thermal conductivity of multilayer graphene oxide (MLGO).

## 2. Materials and Methods

### 2.1. Experimental Materials

The main experimental materials are shown in Table 1. The multilayer graphene oxide (MLGO) was black powder, provided by Suzhou Tanfeng Graphene Technology Co., Ltd. (Suzhou, China). The MLGO had 6~10 layers, each layer was 5~50 μm, the oxygen content was higher than 0.5 wt.% and the sulfur content was lower than 3 wt.%. Polyethylene glycol-400 (PEG-400) was provided by Asia Pacific United Chemical Co., Ltd. (Wuxi, China), with a molecular weight of 380–420, and with an appearance as a colorless transparent viscous

liquid. Sodium Dodecyl Sulfonate (SDS) was supplied by Fuzhou Feijing Biotechnology Co., Ltd. (Fuzhou, China), with a white powder appearance and a purity higher than 99%. Fast-growing poplar wood was supplied by Dehua Bunny Decoration New Material Co., Ltd. (Deqing, China). Poplar veneers were sawed into a size of 150 mm × 150 mm × 2 mm, and the surface was required to be clean and flat without knots and cracks. They were simple sanded with 320-grit sandpaper. Water-soluble melamine formaldehyde resin (MF) was provided by Ningbo Zhongchen Import & Export Co., Ltd. (Ningbo, China), with a solid content of 70% and with an appearance as a colorless transparent viscous liquid. The distilled water was laboratory-made. Submicron multilayer graphene oxide (SMLG) was ball-milled from MLGO under the optimum ball milling process conditions of SDS wet ball milling and a 30 Hz vibration frequency for 60 min.

**Table 1.** Experimental material details.

| Experimental Materials | Source | Details |
|---|---|---|
| Multilayer graphene oxide (MLGO) | Tanfeng Graphene Technology Co., Ltd. (Suzhou, China) | black powder 6~10 layers, 5~50 μm each lamella oxygen content no less than 0.5 wt.% sulfur content lower than 3 wt.% |
| Polyethylene glycol-400 (PEG-400) | Asia Pacific United Chemical Co., Ltd. (Wuxi, China) | colorless transparent viscous liquid molecular weight 380 to 420 |
| Sodium Dodecyl Sulfonate (SDS) | Fuzhou Feijing Biotechnology Co., Ltd. (Fuzhou, China) | white powder |
| Fast-growing poplar wood | Dehua Bunny Decoration New Material Co., Ltd. (Deqing, China) | 150 mm × 150 mm × 2 mm clean and flat surface no knots and cracks simple sanding with 320-grit sandpaper |
| Water-soluble melamine-formaldehyde resin (MF) | Ningbo Zhongchen Import & Export Co., Ltd. (Ningbo, China) | colorless transparent viscous liquid 70% solids |
| Distilled water | Laboratory-made | |
| Submicron multilayer graphene oxide (SMLG) | Laboratory preparation | Ball milled with SDS wet ball milling, 30 Hz for 60 min from MLGO |

### 2.2. Experimental Design

The 3 × 2 × 4 comprehensive factorial experiment was conducted to explore the effect of the ball milling technique on the particle size of multilayer graphene oxide (MLGO). The three influencing factors for ball milling were the environment (dry ball milling (G), PEG-400 wet ball milling (P), SDS wet ball milling (S), the vibration frequency of ball milling (20, 30 Hz) and the time range of ball milling (30, 60, 90 and 120 min). Therefore, 24 graphene oxide dispersion solutions were evaluated in the experiments. The specific ball milling conditions are shown in Table 2.

**Table 2.** Comprehensive factorial test level table.

| Process Condition Number | Ball Milling Environment | Ball Milling Frequency/Hz | Ball Milling Time/min |
|---|---|---|---|
| G-20-30 | | | 30 |
| G-20-60 | | | 60 |
| G-20-90 | | 20 | 90 |
| G-20-120 | | | 120 |
| G-30-30 | dry ball milling | | 30 |
| G-30-60 | | | 60 |
| G-30-90 | | 30 | 90 |
| G-30-120 | | | 120 |

**Table 2.** *Cont.*

| Process Condition Number | Ball Milling Environment | Ball Milling Frequency/Hz | Ball Milling Time/min |
|---|---|---|---|
| P-20-30 | | | 30 |
| P-20-60 | | | 60 |
| P-20-90 | | 20 | 90 |
| P-20-120 | PEG-400 wet ball mill | | 120 |
| P-30-30 | | | 30 |
| P-30-60 | | | 60 |
| P-30-90 | | 30 | 90 |
| P-30-120 | | | 120 |
| S-20-30 | | | 30 |
| S-20-60 | | | 60 |
| S-20-90 | | 20 | 90 |
| S-20-120 | SDS wet ball mill | | 120 |
| S-30-30 | | | 30 |
| S-30-60 | | | 60 |
| S-30-90 | | 30 | 90 |
| S-30-120 | | | 120 |

Except from that, a control experiment was also conducted based on the optimal impregnating agent formula. The graphene-oxide-impregnated veneer without ball milling was compared with the graphene-oxide-impregnated veneer with the best ball milling effect. The effect of ball milling on the thermal conductivity of the impregnated poplar veneer was analyzed.

*2.3. Procedure*

2.3.1. Graphene Oxide Ball Milling

As shown in Figure 1, a grinding jar with a volume of 25 mL was selected to add the ball milling material to less than 16 mL. Clean and dry small steel balls with diameters of 3 mm were used as ball milling aids, and 10 g of small steel balls was added to each grinding jar. The test materials were weighed using an electronic balance (HC UTP-313, Huachao Electric, Shanghai, China), and 1 g of multilayer graphene oxide (MLGO) powder was added to each grinding jar according to a 10:1 ball-to-powder ratio. An amount of 4.04 g of PEG-400 and 4.04 g of distilled water were added to each grinding jar with PEG-400 wet ball milling, and 0.08 g of SDS and 8 g of distilled water were added to each grinding jar with SDS wet ball milling. Ball milling was performed in specific conditions using a ball mill homogenizer (MillMix 20, DOMEL, Zelezniki, Slovenia). After the ball mill material hit the tank and the ball mill material interacted with the tank during the working process of the ball mill homogenizer, including impact, shear and rolling, a large amount of heat energy was generated, which had to be cooled. The tank body was cooled by dry ice every 30 min when the vibration frequency of the ball mill was 20 Hz and every 20 min when the vibration frequency of the ball mill was 30 Hz. The tank was cooled after ball milling and then opened to separate the steel beads from obtaining powdered multilayer graphene oxide (dry ball milling method) or dispersion (wet ball milling method). The samples were diluted with water and ultrasonically dispersed in an ultrasonic cleaner (JP-010S, Jiemeng, Shenzhen, China) for 5 min before testing.

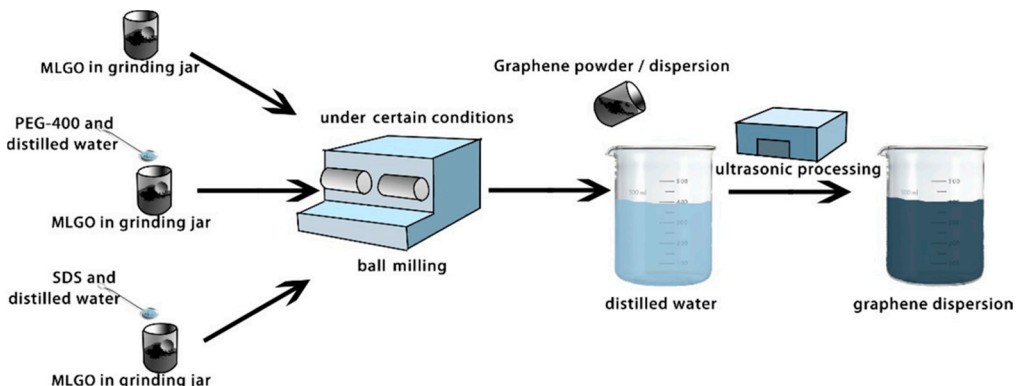

**Figure 1.** Graphene oxide ball milling procedure.

### 2.3.2. Preparation of Modified Veneers

According to our previous work [19], the optimal formulation for graphene oxide dispersion was 50 wt.% MF concentration and 1 wt.% multilayer graphene oxide. The experimental group used a graphene oxide dispersion from the submicron multilayer graphene oxide (SMLG) at an MF concentration of 50 wt.%, SMLG of 1 wt.% and an impregnation solution of 600 g. The SMLG was prepared under the optimum ball milling process conditions of SDS wet ball milling with 30 Hz vibration frequency for 60 min. The control group was set up using the graphene oxide dispersion liquid that was not ball-milled, and the immersion liquid was 600 g according to the MF concentration of 50 wt.% and the non-ball-milled graphene oxide concentration of 1 wt.%. After mixing, they were thoroughly stirred with a glass rod, and two kinds of impregnants were produced after ultrasonic treatment for 20 min.

Before microwaving, the moisture content of veneer was controlled between 10%~13%. All veneers were microwave-treated with 100% radiant power for 50s to improve the permeability of the veneer and were then adiabatically dried at 103 °C. The dried microwave pretreated poplar veneer (MPW) was stacked neatly in the vacuum impregnation equipment. Simultaneously, the veneers were left with gaps and were pressed with weights on top. The laboratory-made vacuum impregnation equipment consisted of an oil-free vacuum pump (SCINCETOOL R410, Sciencetool International, Las Vegas, NV, USA) and a vacuum dryer (PC-3, Fujiwara, Taizhou, China). the impregnation liquid was injected until the liquid level was 2 cm higher than the top of the MPW. The vacuum was then evacuated for 1 h until the −0.1 MPa level was reached and held at normal pressure for 0.5 h. After impregnation, the MPW was rinsed with distilled water, the excess water was wiped away with a paper towel and then the MPW was placed in a constant temperature blast dryer until absolutely dry. Before testing, the dried MPW was placed in a room at 20 °C and 70% relative humidity for two weeks to maintain an even moisture content.

### 2.4. Measurement and Characterization

### 2.4.1. Particle Size Detection

The particle size of graphene oxide with different ball milling conditions was measured by a nano-laser particle sizer (BT-90, Dandong, China). Graphene oxide is hydrophobic and prone to agglomerating and sedimenting in water, which significantly influences particle size detection results. The sample was diluted with water for better dispersion and more accurate particle size detection. Specifically, it was ultrasonically dispersed for more than 5 min in an ultrasonic cleaner. The ball-milled multilayer graphene oxide obtained with the optimal ball-milling process was recorded as submicron multilayer graphene oxide (SMLG).

### 2.4.2. Statistical Analysis

IBM SPSS Statistics performed a single dependent variable multi-factor analysis of variance (ANOVA). In addition, it analyzed the effect of the particle size of multilayer graphene oxide with different ball milling environments, ball milling vibration frequencies and ball milling time treatments. The significance level for all data analyses was set at 5%.

### 2.4.3. Spectral Analysis

The width and position of Raman spectra correlate with their layer number and can characterize the number of graphene oxide layers. All samples were dried prior to testing. The changes in multilayer graphene oxide before and after ball milling were investigated by performing Raman characterization tests. The spectra of the multilayer graphene oxide feedstock sample (MLG) and the submicron multilayer graphene oxide (SMLG) were compared by a laser Raman spectrometer (DXR 532, Thermo, Waltham, MA, USA). Similarly, an infrared spectrometer (VERTEX 80V, Bruker, Karlsruhe, Germany) was used to characterize the samples.

### 2.4.4. X-ray Diffraction Characterization

X-ray diffraction can resolve the crystal structure and composition of materials. A horizontal X-ray diffractometer (Ultima IV, Rigaku, Tokyo, Japan) was used to characterize and compare the spectra of multilayer graphene oxide raw samples (MLG) and submicron multilayer graphene oxide (SMLG). All samples were dried prior to testing. Changes in the structure of multilayer graphene oxide before and after ball milling were analyzed.

### 2.4.5. Thermal Conductivity Determination

Specimens prepared from non-ball-milled graphene oxide/MF impregnation and SMLG/MF impregnation modifications were measured with the steady-state plate method by a thermal conductivity tester (YBF-4, Hangzhou Dahua, Hangzhou, China). The steady state value was set at 100 °C. The thermal conductivity ($\lambda$) was calculated by Formula 1.

$$\lambda = mc \cdot \left. \frac{\Delta T}{\Delta t} \right|_{T=T_2} \cdot \frac{h}{T_1 - T_2} \cdot \frac{1}{\pi R^2} \tag{1}$$

where

$m$ is the mass of the copper plate on the low-temperature side, kg;
$c$ is the specific heat capacity of the copper plate, taken as a constant 385 J·kg$^{-1}$·K$^{-1}$;
$T_1$ is the temperature of the copper plate on the high-temperature side when reaching the steady state, K;
$T_2$ is the temperature of the copper plate on the low-temperature side when reaching the steady state, K;
$\left. \frac{\Delta T}{\Delta t} \right|_{T=T_2}$ is the heat dissipation rate of the copper heat sink at $T_2$, mV·s$^{-1}$;
$h$ is the thickness of the tested sample, m;
$\pi R^2$ is the heat dissipation area of the copper plate on the low-temperature side, m$^2$.

## 3. Results and Discussions

### 3.1. Effects of the Ball Milling Process on Particle Size

Table 3 shows the particle size parameters of multilayer graphene oxide treated in different ball milling environments, vibration frequencies and times. It shows that the ball milling treatment could effectively reduce the particle size of multilayer graphene oxide. The median diameter (D50), commonly used to evaluate the particle size of powders in powder production and applications, was used as the evaluation index. A univariate multi-factor ANOVA was conducted on the ball mill environments, vibration frequencies and times. In terms of the particle size of multilayer graphene oxide, the ball milling environment ($p = 0.001$, $p < 0.01$) had a highly significant effect, the ball milling vibration

frequency ($p = 0.152$, $p > 0.05$) had a widespread impact and the ball milling time ($p = 0.047$, $p < 0.05$) had a significant effect.

**Table 3.** Particle size distribution parameter of graphene oxide after ball milling.

| Process Condition Number | Average Diameter/nm | D50/nm | D [4, 3]/nm | D [3, 2]/nm | Dispersion |
|---|---|---|---|---|---|
| G-20-30 | 1584 | 758 | 720 | 705 | 0.200 |
| G-20-60 | 1552 | 586 | 521 | 519 | 0.324 |
| G-20-90 | 1664 | 362 | 332 | 329 | 0.034 |
| G-20-120 | 849 | 471 | 473 | 469 | 0.411 |
| G-30-30 | 1584 | 503 | 483 | 479 | 0.208 |
| G-30-60 | 1077 | 394 | 380 | 377 | 0.528 |
| G-30-90 | 1171 | 306 | 278 | 274 | 0.539 |
| G-30-120 | 895 | 345 | 332 | 329 | 0.490 |
| P-20-30 | 1426 | 476 | 448 | 443 | 0.315 |
| P-20-60 | 1167 | 249 | 232 | 231 | 0.425 |
| P-20-90 | 962 | 198 | 184 | 183 | 0.579 |
| P-20-120 | 913 | 202 | 187 | 186 | 0.540 |
| P-30-30 | 1552 | 350 | 307 | 303 | 0.385 |
| P-30-60 | 1102 | 234 | 213 | 211 | 0.414 |
| P-30-90 | 964 | 176 | 164 | 163 | 0.537 |
| P-30-120 | 910 | 213 | 195 | 193 | 0.542 |
| S-20-30 | 1099 | 438 | 428 | 425 | 0.460 |
| S-20-60 | 910 | 176 | 165 | 164 | 0.527 |
| S-20-90 | 1117 | 201 | 184 | 183 | 0.568 |
| S-20-120 | 1061 | 194 | 172 | 170 | 0.649 |
| S-30-30 | 1060 | 247 | 230 | 227 | 0.473 |
| S-30-60 | 819 | 124 | 116 | 115 | 0.571 |
| S-30-90 | 1073 | 144 | 130 | 129 | 0.454 |
| S-30-120 | 1053 | 130 | 116 | 115 | 0.596 |

The median diameter of multilayer graphene oxide (MLGO) treated by dry ball milling had a similar trend as that of MLGO treated by PEG-400 wet ball milling. Their median diameter gradually decreased with the increase in ball milling time and reached a minimum value when the ball milling time was 90 min. After that, it showed an apparent increasing trend. This may be explained that the rolling action of the ball milling process reduced the surface morphology and crystallinity of the graphene oxide as time passed. Then, the MLGO powder particles agglomerated again [20–22]. Due to the high-temperature decomposition of PEG 400, the grinding jar gave off a pungent odor when opened. In the SDS wet ball milling environment, the median diameter of the multilayer graphene oxide gradually decreased with the ball milling time. The median particle size reached a minimum value when the ball milling time was 60 min and showed a stable fluctuation trend after that. Compared to the PEG-400 wet ball milling environment, the SDS wet ball milling environment was more effective. This may be explained by the better dispersion of multilayer graphene oxide in the SDS anionic surfactant, which minimized the median diameter in a shorter time [23–25].

### 3.1.1. Influence of Ball Milling Environment on the Median Diameter of Multilayer Graphene Oxide

Figure 2 shows the effect of the ball milling environment on the particle size D(50) of the multilayer graphene oxide after ball milling. The wet ball milling method was better for obtaining graphene oxide with smaller particles than the dry ball milling method. PEG-400 and sodium dodecyl sulfonate were surfactants that significantly reduced the surface tension of water and changed the wettability of the solid surface. This explains the more uniform dispersion of multilayer graphene oxide particles in wet ball milling and the improved efficiency and quality of the ball milling process. Furthermore, compared to PEG-400, multilayer graphene oxide reduced particle size after SDS ball milling more

effectively. In order of preference, the environmental conditions for ball milling were SDS wet ball milling > PEG-400 wet ball milling > dry ball milling.

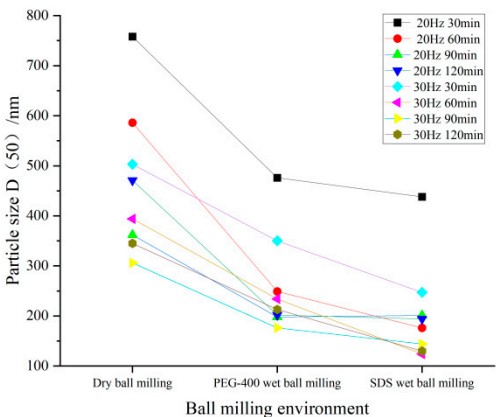

**Figure 2.** Effect of ball milling environment on particle size D(50).

### 3.1.2. Effect of Ball Milling Vibration Frequency on the Median Diameter of Multilayer Graphene Oxide

Figure 3 shows the effect of ball milling vibration frequency on the particle size D(50) of multilayer graphene oxide after ball milling. Multilayer graphene oxide treated with 30 Hz ball milling vibration frequency had a more significant particle size reduction effect than 20 Hz ball milling vibration frequency when the other conditions were fixed. The particle size of multilayer graphene oxide was smaller at higher ball milling vibration frequencies. This may be explained by the increased number of collisions per unit time between the multilayer graphene oxide and the small steel balls assisted by the ball mill.

### 3.1.3. Effect of Ball Milling Time on the Median Diameter of Multilayer Graphene Oxide

Figure 3 shows the effect of ball milling time on the particle size D(50) of multilayer graphene oxide after ball milling. In certain conditions, the average median diameter of multilayer graphene oxide gradually decreased and then increased with the lengthening of the ball-milling time. As the ball milling time became longer, the multilayer graphene oxide collided and sheared with the small steel balls of the ball milling aid more times. It was found that the high local temperatures generated during the collision caused the graphene oxide particles to re-agglomerate.

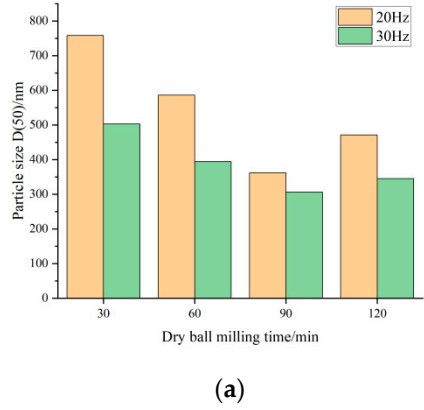

(**a**)

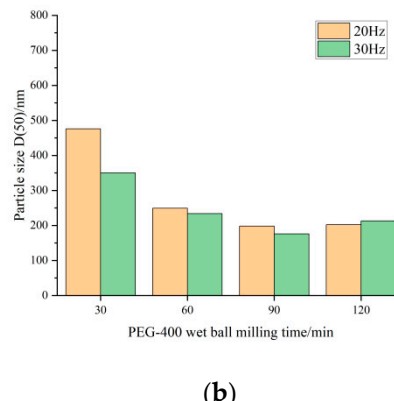

(**b**)

**Figure 3.** *Cont.*

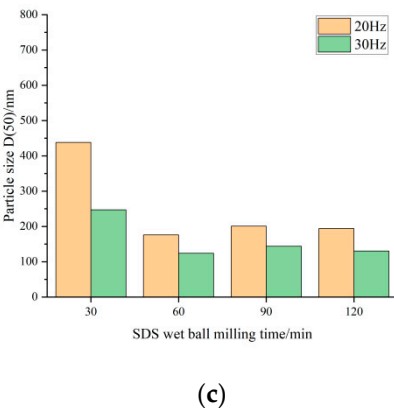

(**c**)

**Figure 3.** Effect of ball milling time and vibration frequency on particle size D(50): (**a**) Dry ball milling; (**b**) PEG-400 wet ball milling; (**c**) SDS wet ball milling.

In summary, the optimal group of ball milling environments was SDS wet ball milling, the ball milling vibration frequency was 30 Hz and the ball milling time was 60 min. There were relatively few large particles in multilayer graphene oxide. Most were small particles, and the size distribution was uniform. The ball-milled multilayer graphene oxide in these process conditions was labeled submicron multilayer graphene oxide (SMLG) for subsequent characterization.

### 3.2. SMLG Characterization Analysis

### 3.2.1. Raman Spectroscopy

Graphene oxide has three distinct Raman features: the D peak near 1350 cm$^{-1}$, the G peak near 1580 cm$^{-1}$ and the 2D peak near 2675 cm$^{-1}$. The D peak is created by defects in the 2D carbon atomic layer and by phonons at the edges of the mediator region. It can reflect defects in the graphene oxide internal structure. Graphene oxide G-peak is caused by the hybridization of sp2 carbon, which is found in the ring of carbon atoms. In addition, the G-peak can define the symmetry and order of the structure. Peak intensity ratios ($I_D/I_G$) are often used to analyze the degree of order in graphene oxide materials. As the ratio becomes higher, the disorder becomes greater and the number of defects increased. The 2D peaks are caused by double resonance jumps of two phonons with opposite momentum in carbon atoms, and their intensity is inversely proportional to the number of graphene oxide layers.

Figure 4 shows that the absorption peak positions of untreated multilayer graphene oxide (MLG) and the optimal multilayer graphene oxide (SMLG) group are nearly identical. The $I_{2D}/I_G$ values of MLG and SMLG were calculated to be 0.367 and 0.54. Additionally, the $I_{2D}/I_G$ value of SMLG was consistent with the value in the Raman spectrum of the four-layer graphene oxide. This means that the SDS wet ball milling process could effectively reduce the number of layers of multilayer graphene oxide to four. The $I_D/I_G$ values of MLG and SMLG were 0.459 and 0.728. This indicated that the carbon atomic layers in graphene oxide decreased, and the defects increased after wet ball milling by SDS.

### 3.2.2. X-ray Diffraction (XRD) Analysis

As shown in Figure 5, SMLG and MLG showed characteristic peaks (002) at around 26.5°. The diffraction intensity of MLG was three times higher than that of SMLG. This indicated that the crystallinity of the SDS wet-ball-milling-treated multilayer graphene oxide decreased, and defects arose in the ordered stacked carbon atomic lamellae. The results were consistent with the Raman spectra analysis [26]. Compared with MLG, the diffraction peak of SMLG at about 26.5° shifted from 26.487° to 26.449°. According to the Bragg equation, it was inferred that the interlayer spacing of MLG was 0.3361 nm and that of SMLG was 0.3366 nm. The interlayer spacing of the multilayer graphene oxide increased slightly.

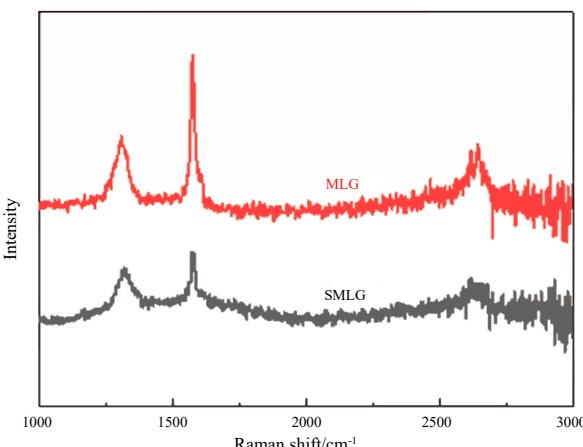

**Figure 4.** Raman image of SMLG and MLG.

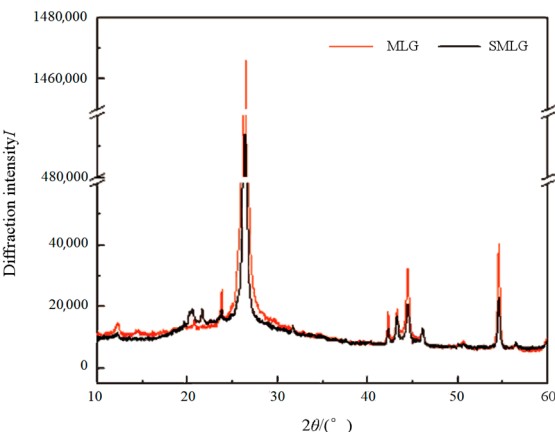

**Figure 5.** XRD image of SMLG and MLG.

### 3.2.3. Fourier Transform Infrared Spectroscopy (FT-IR) Analysis

As shown in Figure 6, both MLG and SMLG had strong and broad absorption peaks around 3400 cm$^{-1}$, which were the -OH stretching vibration peaks, among the absorption peaks, that of of SMLG was more substantial around 3400 cm$^{-1}$. The absorption peak of SMLG at 1700 cm$^{-1}$ was significantly weakened, which was the C=O stretching vibration peak [27,28]. The absorption peak at 1630 cm$^{-1}$ originated from the water molecules adsorbed on the surface of the GO structure, which meant the bending vibration of C-OH. In addition, there was no significant difference between MLG and SMLG at 1630 cm$^{-1}$. The vibration absorption peaks of C-H and C-OH were at 1470 cm$^{-1}$ and 1370 cm$^{-1}$, respectively, and the absorption peaks of SMLG at those two places almost disappeared. These changes indicated that oxygen-containing functional groups were introduced during the preparation of multilayer graphene oxide by SDS wet ball milling. These oxygen-containing functional groups may have come from strongly adsorbed water molecules that were not removed during the drying process. In addition, the multilayer graphene may have been oxidized by air during the ball milling process. In addition, the C-C bond was broken, the bonding force was reduced and the oxygen-containing functional groups were introduced between the carbon layers. During the ball milling process, the carbon-oxygen double bond was opened, the C=O was reduced to C-O and the hydrogen atom on C-H was replaced with a hydroxyl group to become C-OH.

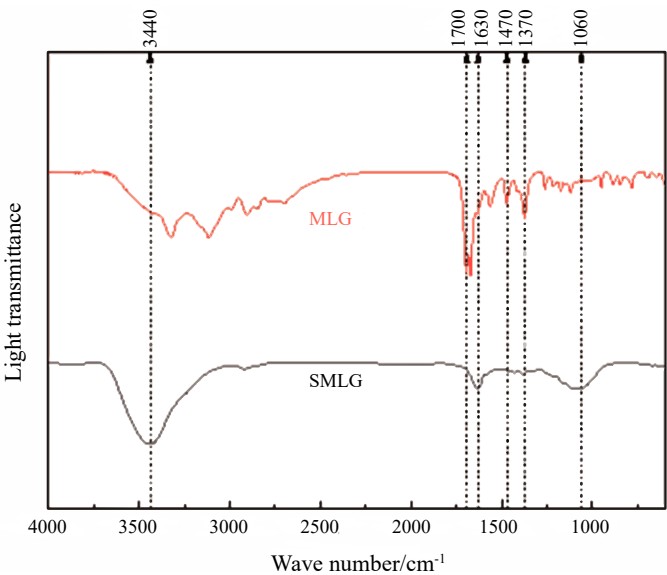

**Figure 6.** FT-IR image of SMLG and MLG.

*3.3. Comparative Analysis of the Thermal Conductivity of Veneer by Ball Milling*

The thermal conductivity of MPW was measured at 0.093 W·m-1·K$^{-1}$. The thermal conductivity of the non-ball-milled graphene oxide impregnated veneer, as shown in Table 4, was measured at 0.174 W·m$^{-1}$·K$^{-1}$, which was an improvement of 87% compared to the untreated material. The thermal conductivity of the veneer impregnated with SMLG was 0.405 W·m$^{-1}$·K$^{-1}$, which was an increase of 335% compared to the untreated material. The ball-milled graphene oxide impregnation solution obviously increased the impregnated veneer's thermal conductivity. This may be explained by the graphene oxide's smaller particle size, which allowed it to fill the wood's cellular spaces during the impregnation process. The gaps in the wood were filled more, and the heat transfer paths were better formed. In conclusion, it demonstrated that the ball milling process played an important role in reducing the particle size of graphene oxide, increasing the efficiency of impregnation and improving the thermal conductivity of the wood [29,30].

**Table 4.** Thermal conductivity of graphene oxide/MF-impregnated veneer.

| Numbering | MF Concentration | SMLG Concentration | Thermal Conductivity λ (W·m$^{-1}$·K$^{-1}$) | The Rate of Increase in λ (%) |
|---|---|---|---|---|
| | | | Average | Compared to Untreated Wood |
| 1-non-ball-milled 2-SMLG | 50% | 1% | 0.174 0.405 | 87 335 |

**4. Conclusions**

The ball milling method is a cost-effective way to obtain smaller particle-sized graphene oxide, providing a theoretical basis for graphene-oxide-modified wood. Using the ball-milled graphene oxide mixed with MF to impregnate poplar veneer, a thermal conductivity wood composite was prepared to improve thermal conductivity in wood and for its application in floor heating. The following are the main conclusions of this study:

(1) The median size reduction of the multilayer graphene oxide after ball milling was in the following order: SDS wet ball milling > PEG-400 wet ball milling > dry ball milling. The particle size reduction of multilayer graphene oxide was more evident at higher ball milling frequencies. With the extension of ball milling time, the median

size of the multilayer graphene oxide first gradually decreased and then showed a fluctuating trend within a specific range.

(2) The optimum ball milling process for reducing the particle size of multilayer graphene oxide was SDS wet ball milling with a ball milling vibration frequency of 30 Hz and a ball milling time of 60 min, and the median diameter of multilayer graphene oxide produced by this ball milling process could be reduced to 124 nm. However, the graphene oxide treated with SDS wet ball milling process had fewer ordered layers of carbon atoms and more defects.

(3) The thermal conductivity of the veneer impregnated with SMLG reached 0.405, and it increased to 335% compared to untreated wood and 132% compared to non-ball-milled graphene-oxide-treated wood. The ball milling process enhanced the thermal conductivity of the wood, providing a theoretical basis for using modified wood in underfloor heating applications.

**Author Contributions:** Conceptualization, W.X.; methodology, Y.M. and N.Z.; software, Y.M. and N.Z.; validation, Y.M., S.W. and N.Z.; formal analysis, Y.M. and N.Z.; investigation, N.Z.; resources, W.X., Y.M. and N.Z.; data curation, Y.M. and N.Z.; writing—original draft preparation, N.Z.; writing—review and editing, S.W. and N.Z. All authors have read and agreed to the published version of the manuscript.

**Funding:** This work was supported by the project from the International Cooperation Joint Laboratory for the Production, Education, Research and Application of Ecological Health Care on Home Furnishing; the Ministry of Education Industry-University Cooperation Collaborative Education Project (202101148004).

**Data Availability Statement:** The data presented in this study are available on request from the corresponding author.

**Acknowledgments:** The authors thank the college of Furnishings and Industrial Design of Nanjing Forestry University for supplying the laboratories and equipment for this experiment.

**Conflicts of Interest:** The authors declare no conflict of interest.

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
