# Peer review of "Effects of the Ball Milling Process on the Particle Size of Graphene Oxide and Its Application in Enhancing the Thermal Conductivity of Wood"

_forests, doi:10.3390/f13081325_

Round 1

Reviewer 1 Report

Graphene and graphene oxide (GO) are different materials with very different properties, including different thermal condustivity (the object of this study). Please revise the title to ensure it is clear that you discuss graphene oxide. Please mention in the introduction the difference between graphene and graphene oxide. Please make it clear where literature that you cite is about graphene nanoparticles, this is different from the graphene oxide you are studying and the reasons for chosing to study GO.

Table 1 in the method section is unlcear due to many lines of text which are not easily matched with the products being described. Please add an extra line between each material to ensure that data for GO, PEG, SDS, wood and MF resins are not confused. This table does not include the SMLG graphene oxide which is mentioned on page 4 in line 13o. Should it?

SMLG is introduced poorly in line 130. Is this the optimal product from the ball milling experiment? If so, please state clearly which type of ball milling and which milling conditions were used to create it.Note - i see that you state this in line 253, however this is too far after the first mention of SMLG in line 130, please work our how to give this information more prominenlty and nearer the first mention of the acronym.

As the SMLG is from the ball milling experiment, please also explain how it is prepared from the dispersion of GO generated at the end of the experiment (Figure 1) as this is a very low dilution, but will need to be concentrated (and possibly cleaned to remove PEG and /or SDS) before it can be mixed with the resin.

In reporting the concentration data for the MF resin used to impregnate the wood you do not indicate whether the resin concentration (50%) is on a weight basis, resin solids basis or volume basis. Please clarify. You also do not make clear whether the 1% GO particles added are 1% by weight relative to the weight of resin solution, or as a solution by volume relative to volume of resin etc etc. There are many other ways this information could be misinterpreted, so please state more clearly the actual basis for this percentage in lines 131-136.

In lines 165-173 you describe both the Raman< FTIR and XRD equipment. Please ammend the heading for line 165 to reflect the different spectroscopic methods being used. Also, please state whether the samples were dried, cleaned or presented to the different techniques in solution state, or other preparation details.

Were the veneers wet or dry or air dry (i.e. at some specific moisture content) when you microwave irradeiated them? Please state.

Did you make any mesurements of the weight of resin taken up by the wood veneers?

In section 2.4.1 you say that only the MLG and the SMLG were subjected to particle size analysis, but in table 3 it is clear that this was done for all ball milled samples. Please revise the text to reflect what you actually did.

In lines 312-314 you mention the thermal conductivity of the treated wood but do not give the value for the untreated wood. please add this to the text here. Presumably this is 93W/m-K based on the % values given but it makes sense to state it to avoid misunderstandings.

The proposed explanation of the method of action given in line 316-317 seems very simple. I suggest you are right with the better accessibility part of the statement. However the question of whether the GO particles have excluded any air is complete speculation and should be removed. Even in non-nanoparticle filled melamine resins it is unusual for air to be excluded. This part should be deleted. Is the mechanism not more related to the good distribution of the GO particles, which could allow them to conduct heat more effectively through the wood as they might form a more nearly continuous chain of particles between one surface of the wood and the other? However this must take into account that this work has not investigated their distribution, or their location relative to cell lumena and cell wall regions.

Line 325-326 you say 'wood thermal conductivity composite', this seems an odd term. would it be better to say 'thermally conductive wood composite'?

Reviewer 2 Report

The authors explore adding graphene to melamine formaldehyde resins used to treat wood/wood composites. The goal of this treatment is to increase the thermal conductivity of wood flooring so that it can be used with radiant heat. 

The introduction is well written and presents relevant previous research.

Table 1 is very confusing. It is split over 2 pages. Also, the columns do not line up and it is hard to read. Section 2.1 has no text but just a table. I recommend changing Table 1 into a paragraph about the materials used.

Lines 100-102: Please give more details about the control experiment.

Figure 1: labels have MLG. But the paper uses MLGO as the acronym.

Line 128: add citation to your previous work.

Line 130: Why is "best ball mills group" abbreviated as SMLG? This needs further explanation. What is SMLG?? (edit: I see this is defined later on in the paper- more description is needed here)

Line 186: What were T1 & T2 in your thermal conductivity plate measurements? What range were you taking these thermal conductivity measurements in?

Figure 2: Is this the average of all dry ball milled samples, etc? All samples that start with the letter G? How can you average these since you have changed lots of other variables? Please be more explicit about what you are plotting.

Same with Figures 3 and 4? Are you grouping the other variables in these averages? If so, that does not seem like an appropriate way to plot these data trends since you're combining other variables. Please give more details, and replot if necessary.

Line 250: Why 60 minutes? Your data (Figure 4) show even smaller particle sizes for 90 minutes and 120 minutes. You claim that 60 minutes is optimal. You should probably soften language here since it is a subjective choice for further study. 

Line 315 "significantly increased the impregnated veneer". If you are claiming statistical significance, you need a p-value here. If you don't have enough replicates to get statistics... (did you only do one sample?) then you can't use definitive language in this interpretation.

Round 2

Reviewer 2 Report

Thank you for addressing my previous comments.